

# Comparative phosphoproteome analysis to identify candidate phosphoproteins involved in blue light-induced brown film formation in *Lentinula edodes*

Tingting Song, Yingyue Shen, Qunli Jin, Weilin Feng, Lijun Fan and Weiming Cai

Institute of Horticulture, Zhejiang Academy of Agricultural Sciences, Hangzhou, China

## ABSTRACT

Light plays an important role in the growth and differentiation of *Lentinula edodes* mycelia, and mycelial morphology is influenced by light wavelengths. The blue light-induced formation of brown film on the vegetative mycelial tissues of *L. edodes* is an important process. However, the mechanisms of *L. edodes*' brown film formation, as induced by blue light, are still unclear. Using a high-resolution liquid chromatography-tandem mass spectrometry integrated with a highly sensitive immune-affinity antibody method, phosphoproteomes of *L. edodes* mycelia under red- and blue-light conditions were analyzed. A total of 11,224 phosphorylation sites were identified on 2,786 proteins, of which 9,243 sites on 2,579 proteins contained quantitative information. In total, 475 sites were up-regulated and 349 sites were down-regulated in the blue vs red group. To characterize the differentially phosphorylated proteins, systematic bioinformatics analyses, including gene ontology annotations, domain annotations, subcellular localizations, and Kyoto Encyclopedia of Genes and Genomes pathway annotations, were performed. These differentially phosphorylated proteins were correlated with light signal transduction, cell wall degradation, and melanogenesis, suggesting that these processes are involved in the formation of the brown film. Our study provides new insights into the molecular mechanisms of the blue light-induced brown film formation at the post-translational modification level.

## INTRODUCTION

*Lentinula edodes*, also known as shiitake mushroom, belonging to Lentinus, is a valuable medicinal and edible fungus (*Ozcelik & Peksen, 2007*). It is a popular edible mushroom and the third most cultivated mushroom in the world (*Philippoussis, Zervakis & Griensven, 2000*). During cultivation, there are at least four growth stages: vegetative mycelial growth with growth substrate colonization, the light-induced brown film formation, primordial formation, and fruiting body development (*Aleksandrova et al., 1998*). The brown film formation on the surface of mature mycelia usually appears on the fruiting body primordia and may represent a speciation step (*Aleksandrova et al., 1998*; *Chum et al., 2008*;

Corresponding author
Weiming Cai, caiwm0527@126.com

*Tsivileva et al., 2005*). In addition, the mycelial surface does not form a brown film, which is easily occupied by pathogenic organisms, such as bacteria, green molds and fungi (*Koo, Lee & Lee, 2013*). Light signals are essential factors in the formation of brown films (*Tang et al., 2013*; *Yin et al., 2017*; *Zhang et al., 2015*). The basic genetic regulatory mechanisms of brown film formation and the influence of environmental factors, especially light, remain unclear. Comparative transcriptome studies revealed that the mechanisms of light-induced brown film formation are related to photosensitivity, signal transduction pathways, and melanin deposition (*Tang et al., 2013*). Several gene ontology (GO) classifications related to brown film formation were revealed by two-dimensional electrophoresis combined with the matrix-assisted laser desorption/ionization tandem time-of-flight mass spectrometry approach and included small molecule metabolic processes, response to oxidative stress, and organic substance catabolic processes (*Tang et al., 2016*). *Kim et al. (2020)* compared the morphological changes and gene expression of *Lentinus edodes* under blue light and continuous dark conditions. Their results indicated that the differential genes were involved in the morphological development of primordia and embryonic muscle, cell adhesion and the structure of cellulose and non-cellulose cell walls that affect the development of fruiting bodies, as well as photoreceptors of blue light signals for fruiting body development and pigment formation (*Kim et al., 2020*). Blue light is an important environmental factor in inducing primordial differentiation and the fruiting body development of mushrooms, such as *Hypsizygus marmoreus*, *Pleurotus ostreatus*, and *Coprinus cinereus* (*Kues et al., 1998*; *Terashima et al., 2005*; *Xie et al., 2018*).

During the growth and development of fungi, the influence of light is very important, and it is also necessary for their growth and development (*Crosson, Rajagopal & Moffat, 2003*). As an external signal, light regulates mycelial growth, primordial differentiation, fruiting body formation, gene expression, and metabolite and enzyme activities through complex light-sensing systems (*Cohen et al., 2013*; *Miyake et al., 2005*; *Wu et al., 2013*; *Zhang et al., 2013*). At least 100 kinds of fungi have light-perception systems, including red, blue, green, and near-violet (*Casas-Flores et al., 2006*). Photoreceptors are proteins that harvest light and produce signals that are then transported to the nucleus to activate the transcription of light-responsive genes (*Hurley et al., 2012*). The white collar-1/white collar-2 (WC-1/WC-2) complex is the main blue-light sensor in *Neurospora crassa*, a model organism for studying photoperiod (*Dunlap, 2006*; *Linden, 1997*). Other blue-light receptors have been successfully identified and cloned, such as the dst1 and dst2 genes in *C. cinereus*, phrA and phrB in *L. edodes*, Cmwc-1 in different strains of *Cordyceps militaris*, and Slwc-1 from *Sparassis latifolia* (*Kuratani et al., 2010*; *Sano et al., 2009*; *Sano et al., 2007*; *Terashima et al., 2005*; *Yang et al., 2017*; *Yang et al., 2012*). However, the molecular mechanisms of blue light-induced brown film formation are still unknown.

With the determinations and in-depth analyses of genome and transcriptome sequences of model organisms, such as *Arabidopsis thaliana*, researchers have realized that it is impossible to understand the functions of organisms from only a gene-based perspective (*Abbott, 2001*). Proteomics studies the compositions, expressions, structures, functions, interactions between proteins and their activities (*Graves & Haystead, 2002*). Isobaric tags for relative and absolute quantification/tandem mass tag (iTRAQ/TMT)-labeling

combined with tandem mass spectrometry is a high-throughput quantitative proteomics application technology developed in recent years (*Zhan et al., 2019*). Compared with relatively stable genomes, proteins are diverse and changeable. In addition, the presence of post-translational modifications (PTMs) and protein processing, such as phosphorylation, glycosylation, and acetylation, are not comparable at the genome or RNA level (*Piehler, 2005*). Proteomics research is a cutting-edge technique in the edible fungi industry. With the effects of abiotic stresses on protein expression levels have been studied the most (*Hernandez-Macedo et al., 2002*; *Liang et al., 2007*).

In this study, an immunoaffinity analysis combined with high-resolution liquid chromatography-tandem mass spectrometry (LC-MS/MS) was used to study the global phosphorylated proteome of brown films induced by blue light. This study provides new insights into the molecular mechanisms of blue light-induced brown film formation at the PTM level.

## MATERIAL AND METHODS

### Materials treatment and protein extraction

The *L. edodes* strain L901 which is a new hybrid strain was obtained from the Zhejiang Academy of Agricultural Sciences. Fungal mycelia were grown at 22 °C under red- and blue-light conditions (LED light sources) for 22 d. The light intensity approximately 100 lux and the incubator illuminated all day. Fungal mycelia were grown were grown in the potato dextrose agar media. Samplings were taken after mycelial changed colour under blue light conditions. The determination of total polysaccharides was performed according to Zhang's description (*Zhang et al., 2018*). For protein extraction, a proper amount of sample was ground in liquid nitrogen into a cellular powder and then transferred to a 5-mL centrifuge tube. The samples were treated with four volumes of lysis buffer (10 mM dithiothreotol, 1% protease inhibitor, and 1% phosphatase inhibitor) and then sonicated three times. The supernatant was centrifuged for 10 min at 4 °C and 5,500 g with an equal volume of Tris equilibrium phenol. The supernatant was taken and precipitated overnight with a fivefold volume of 0.1 M ammonium acetate/methanol. The protein precipitation was washed sequentially with methanol and acetone. The protein was redissolved in 8 M urea, and the protein concentration was determined using a bicinchoninic acid assay kit (P0012, Beyotime, Shanghai, China) according to the manufacturer's instructions.

### Trypsin digestion, TMT labeling, and HPLC fractionation

For digestion, the final concentration of dithiothreotol in the protein solution was 5 mM and was reduced at 56 °C for 30 min. The 11-m final concentration of iodoacetamide was incubated at room temperature for 15 min. Finally, the urea concentration of the sample was diluted to less than 2 M. Trypsin was added at 1:50 trypsin-to-protein mass ratio for the first digestion overnight and 1:100 trypsin-to-protein mass ratio for a second 4 h-digestion.The trypsinase-hydrolyzed peptide segments were desalted using a Strata X C18 (Phenomenex) and then freeze-dried in a vacuum. The peptide segment was dissolved in 0.5 M Triethylammonium bicarbonate and labeled according to the instructions of the TMT kit (90066,Thermo-Scientific, Rockford, IL, USA). The simple operation was as

follows: the labeled reagent was dissolved in acetonitrile after thawing, incubated at room temperature for 2 h after mixing with the peptide segment, desalinated after mixing with the labeled peptide segment, and freeze-dried in a vacuum.

The tryptic peptides were fractionated using high pH reverse-phase HPLC on an Agilent 300Extend C18 column (5-$\mu$m particles, 4.6-mm ID, 250-mm length). Briefly, peptides were first separated using a gradient of 8% to 32% acetonitrile (pH 9.0) over 60 min into 60 fractions. Then, the peptides were combined into six fractions and dried by vacuum centrifugation.

## Affinity enrichment

Peptide mixtures were first incubated with an immobilized metal ion affinity chromatography (IMAC) microsphere suspension and vibrated in loading buffer (50% acetonitrile and 6% trifluoroacetic acid). IMAC microspheres were used Ti. The IMAC microspheres enriched with phosphopeptides were collected by centrifugation, and the supernatant was removed. To remove nonspecifically adsorbed peptides, the IMAC microspheres were washed with loading buffer and 30% acetonitrile plus 0.1% trifluoroacetic acid, sequentially. To elute the enriched phosphopeptides from the IMAC microspheres, elution buffer containing 10% $NH_4OH$ was added, and the enriched phosphopeptides were eluted with vibration. The resulting peptides were desalted with C18 ZipTips (Millipore) and lyophilized for the LC-MS/MS analysis.

## LC-MS/MS analysis

The tryptic peptides were dissolved in 0.1% formic acid and directly loaded onto a home-made reversed-phase analytical column (15-cm length and 75-$\mu$m i.d.). The gradient increased from 6% to 23% solvent B (0.1% formic acid in 98% acetonitrile) over 26 min, 23% to 35% in 8 min and to 80% in 3 min. It was then held at 80% for the last 3 min, at a constant flow rate of 400 nL/min on an EASY-nLC 1000 UPLC system.

The peptides were subjected to an NSI source followed by MS/MS in Q ExactiveTM Plus (Thermo) coupled online to the UPLC. The electrospray voltage applied was 2.0 kV. The m/z scan range was 350 to 1,800 for a full scan, and intact peptides were detected in the Orbitrap at a resolution of 70,000. Peptides were then selected for MS/MS using normalized collision energy (NCE) set as 28, and the fragments were detected in the Orbitrap at a resolution of 17,500. The data-dependent procedure alternated between one MS scan and 20 MS/MS scans with a 15.0-s dynamic exclusion. The automatic gain control was set at 5E4. The fixed first mass was set as 100 m/z.

## Database search

The MS data were retrieved using Maxquant (v1.5.2.8) using the following search parameter settings: the database was Lentinula_edodes_uniprot (https://www.uniprot.org/proteomes/?query=organism:5353&sort=score); an anti-database was added to calculate the false positive rate (FDR) caused by random matching and a common contamination library was added to eliminate the contamination proteins from the results. Trypsin/P was specified as the cleavage enzyme, allowing up to four missing cleavage events. The mass tolerance for precursor ions was set as 20 ppm in the first search and 5 ppm in the main search,

and the mass tolerance for fragment ions was set as 0.02 Da. Carbamidomethyl on Cys was specified as a fixed modification, and acetylation and oxidation of Met were specified as variable modifications. The FDR was adjusted to <1%, and the minimum score for modified peptides was set >40.

## Annotation methods and functional enrichment

The GO annotation on the proteomics level was derived from the UniProt-GOA database (http://www.ebi.ac.uk/GOA/). First, the system converted the protein ID to UniProt ID, matched the GO ID with the UniProt ID, and then extracted the corresponding information from the UniProt-GOA database based on the GO ID. If there was no protein information queried in the UniProt-GOA database, then algorithm software based on the protein sequence, InterProScan, was used to predict the GO function of the protein.

The KEGG database was used to annotate protein pathways. First, the KEGG online service tool KAAS was used to annotate the submitted proteins, and then KEGG mapper was used to place the annotated proteins into the corresponding pathways in the database. WoLF PSORT, a software for predicting subcellular localization, was used to annotate the submitted proteins for subcellular localization. Fisher's exact test was used to detect differentially modified proteins against the background of identified proteins. A $P$-value of less than 0.05 was considered significant. The softwares motif-x and MoMo were used to analyze the models of sequences that contained the amino acids in specific positions of modified 13-mers (six amino acids upstream and downstream of the site) in all the protein sequences.

## RESULTS

### Characteristics of quantitative phosphoproteomic data in *L. edodes* mycelia

Using affinity enrichment followed by LC-MS/MS, the phosphoproteomic changes in *L. edodes* mycelia grown in red or blue light were investigated. A flow chart of our experiment is exhibited in Fig. 1A. Pearson's correlation coefficient between the two groups showed sufficient reproducibility (Fig. 1B). In this study, 160,949 secondary spectra were obtained by MS analyses. After searching the theoretical protein data, the effective number of spectra was 22,857 and the utilization rate of the spectra was 14.2%. In total, 8,830 peptides and 7,777 phosphorylated peptides were identified. There were 11,224 phosphorylation modification sites on 2,786 proteins, of which 9,243 sites on 2,579 proteins provided quantitative information (Fig. 1C). The first-order mass errors of most spectra are less than 10 ppm, which is in accordance with the high accuracy of the MS (Fig. 1D). Most of the peptides were distributed in 7–20 amino acids, which was in accordance with the general rules of trypsin-based enzymatic hydrolysis and high energy collision dissociation (HCD) fragmentation, indicating that the sample preparation and the quality accuracy of the mass spectrometer reached the standard required (Fig. 1E). The detailed information regarding the identified peptides are listed in Table S1.

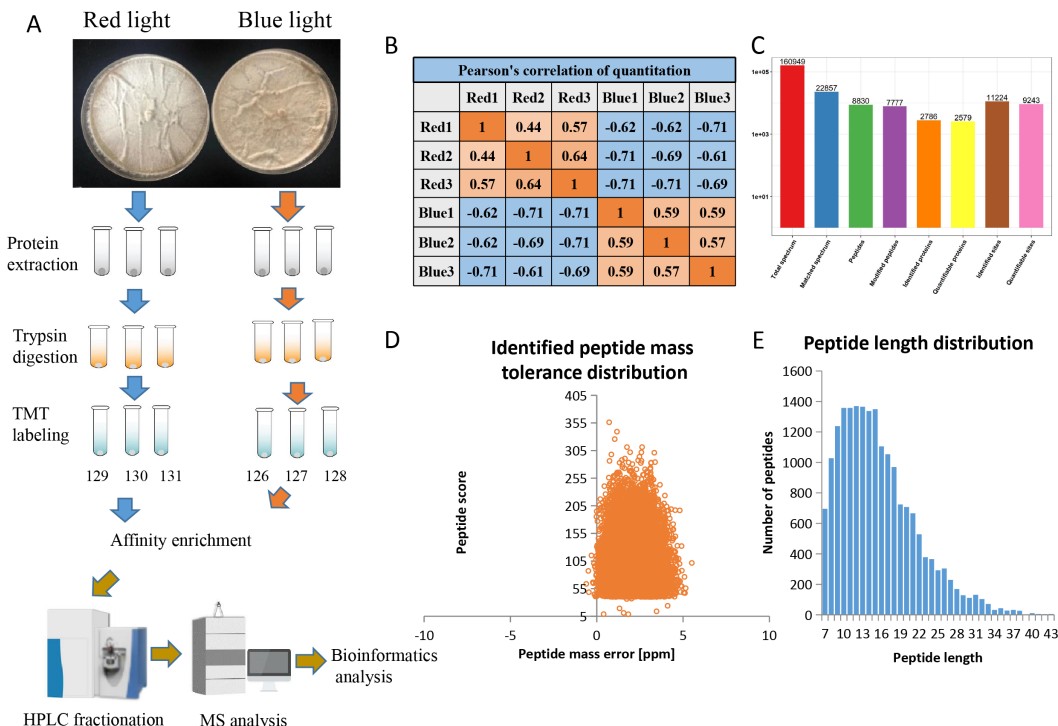

**Figure 1** **Overview of the phosphorylation proteomes.** (A) The pictures showed the fungal mycelia under different illumination for 22 days. Experimental strategy for the quantitative analysis of phosphorylation proteomes from red and blue light treatment groups. (B) Pearson's correlation of the phosphorylation proteomes from two sample groups (three biological replicates for each group). (C) Basic statistical data of MS results. (D) Mass error distribution of all identified phosphorylated peptides. *X*-axis: Peptide Score; *Y*-axis: Peptides mass delta. (E) Length distribution of all identified phosphorylated peptides. *X*-axis: No. of Peptide; *Y*-axis: Peptide length.

## Analysis of phosphorylation sites

In *L. edodes* mycelia, 977 (35.07%) phosphoproteins were modified at a single site, 519 (18.63%) at two sites, and 1,290 (46.3%) at three or more phosphosites (Fig. 2A). Interestingly, some proteins contained a large number of phosphosites. For example, there are 34 phosphosites in a non-specific serine/threonine protein kinase (A0A1Q3E061), 45 phosphosites in a regulatory transcript from a polymerase II promoter-related protein (A0A1Q3ERS8) and 53 phosphosites in a SRC Homology 3 (Sh3) domain-containing protein (A0A1Q3ENM7) (Table S1).

To analyze the density levels of the phosphorylation sites in each protein, the phosphorylated proteome of *L. edodes* was compared with those of other species. The average number of phosphorylation sites per protein in *L. edodes* is 3.22, which is similar to the numbers in Bombyx mori (3.07), Nicotiana tabacum (3.05), and Physcomitrella patens (3.44) (Fig. 2B) (*Fang et al., 2016*; *Lu et al., 2019*; *Shobahah et al., 2017*).

## Characteristics of the identified phosphoproteins in *L. edodes*

To predict the possible functions of the identified phosphoproteins, a GO classification analysis was performed. Most of the proteins were classified into three GO categories
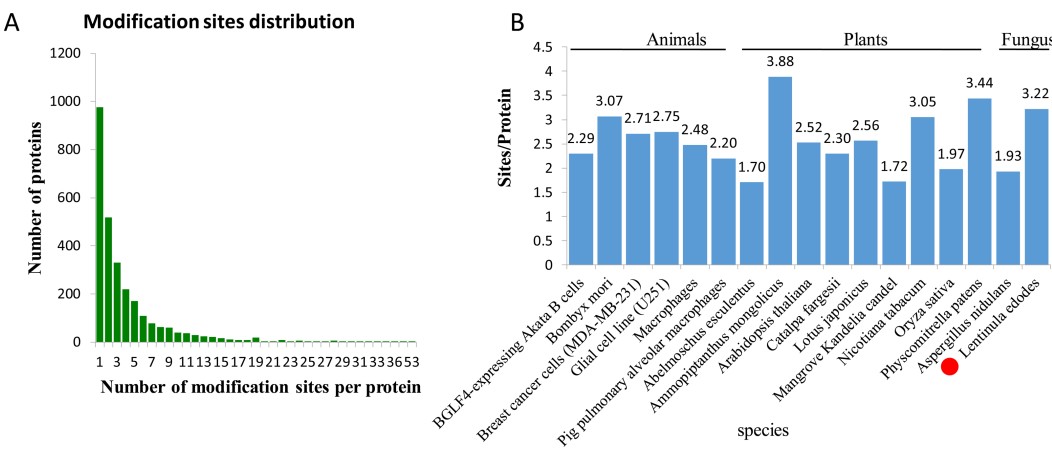

**Figure 2** **Analysis of the density of phosphorylation sites.** (A) Modification phosphorylated sites distribution of all identified peptides. (B) Comparison of the average densities of phosphorylation sites per protein among various species.

(Fig. 3A). Specifically, 594 proteins were annotated as 'metabolic process', 519 proteins were annotated as 'cellular process', and 361 proteins were annotated as 'single-organism process'. In the cellular component category, the largest terms were 'cell' (289 proteins), 'organelle' (186 proteins), and 'macromolecular complex' (154 proteins). In the molecular function category, 'binding' (846 proteins), 'catalytic activity' (627 proteins), and 'transporter activity' (51 proteins) were the three top dominant terms. The euKaryotic Ortholog Groups annotation clustered all the phosphoproteins into four major categories. The 'cellular processes and signaling' category contained the largest number of proteins (Fig. 3B). Most identified phosphoproteins were grouped into 13 subcellular component categories predicted by WoLF PSORT software, including 783 nuclear, 380 cytoplasmic, and 275 mitochondrial proteins (Fig. 3C). The detailed annotation information for all the identified phosphoproteins are listed in Table S2.

## Protein motifs associated with phosphorylation

Among the identified phosphosites in *L. edodes*, 8,645 sites occurred at serine residues, 2239 sites at threonine residues, and 340 sites at tyrosine residues (Fig. 4A). To understand the upstream pathway of the identified phosphorylated proteins, a motif analysis was carried out using MOMO and Motif-X software. A number of conserved phosphorylation motifs were enriched in the phosphorylated proteins of *L. edodes* (Table S3). A total of 7,741 distinct sequences containing 13 residues were obtained, with 6 upstream and 6 downstream residues around each phosphosite (Table S4). The five S-based motifs containing the largest numbers of sequences were 'sP', 'RxxsP','PxsP''Gs', and 'RRxS', and the five top T-based motifs were 'tP','tPP','RxxtP', RxtP', and 'Rxxt'. A Y-based motif, 'Rxxxxxy', was identified. Two position-specific heat maps of upstream and downstream amino acids at all the identified phosphorylated serine or threonine sites. For the S-based motifs, strong preferences for glutamic acid, lysine, and arginine upstream, and aspartic acid, glutamic acid, and proline downstream, of the phosphorylation sites were observed.
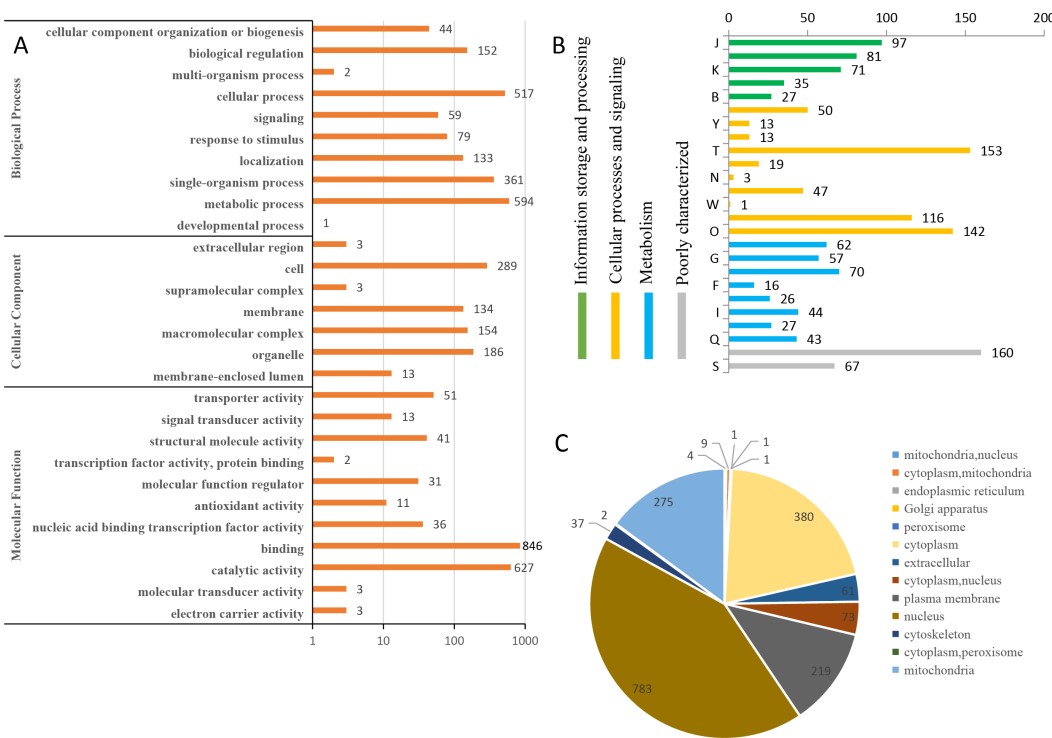

**Figure 3** **Annotation and classification of all identified phosphorylated proteins.** (A) GO analysis of all phosphorylated proteins. All proteins were classified by GO terms based on their biological process, cellular component and molecular function. (B) The euKaryotic Ortholog Groups annotation clustered all the phosphoproteins into four major categories: Information storage and processing, Cellular processes and signaling, Metabolism and Poorly characterized.(C) Subcellular locations ofall identified phosphorylated proteins.

For the T-based motifs, preferences for lysine, proline, and arginine upstream, and aspartic acid and proline downstream, of the phosphorylation sites were observed (Fig. 4C).

## Differentially phosphorylated proteins (DPPs) in response to a blue-light treatment

To compare the DPPs between red- and blue-light treated samples, expression profiles of the proteins generated by MeV software are shown in a heatmap (Fig. 5A). The screening of DPPs followed the following criteria: change threshold ≥1.5 times and $t$-test $p$-value <0.05. Among these DPPs, 475 sites in 317 phosphorylated proteins were up-regulated and 349 sites in 243 phosphorylated proteins were down-regulated (Fig. 5B and Table S5). Based on the subcellular localizations predicted by WoLF PSORT software, all the DPPs were classified into 10 subcellular components. There were 204 nuclear localized DPPs, 82 cytoplasmic localized DPPs, and 51 plasma membrane localized DPPs (Fig. 5C).

## Functional enrichment analysis of the DPPs

To understand the biological functions of these phosphorylated proteins, GO, KEGG and protein domain enrichment analyses of DPPs were carried out. For biological process, cellular component, and molecular function categories, the DPPs were mostly enriched in

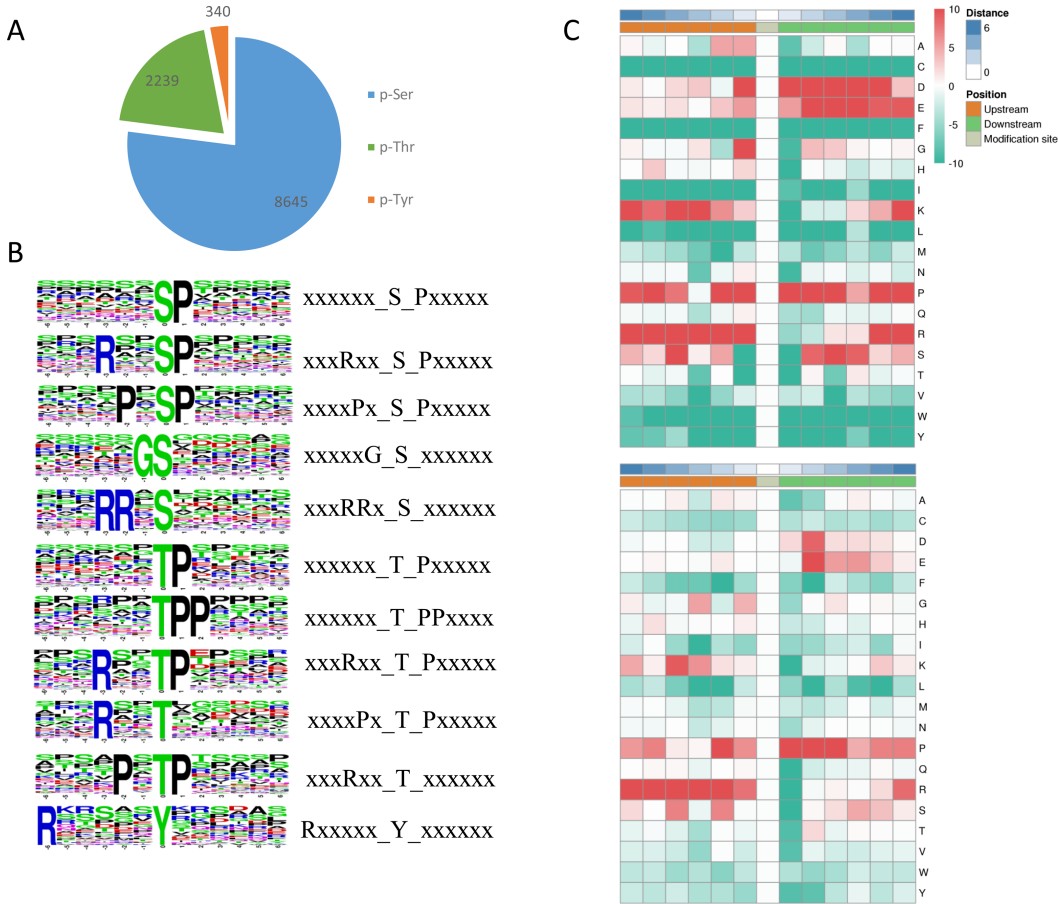

**Figure 4 Phosphosite types and peptide motifs associated with phosphorylation.** (A) The distribution of phosphosites between serine, threonine and tyrosine residues. (B) Motif analysis of the amino acids surrounding the phosphosites. Sequence logo representation of 5 S-based and 5 T-based conserved phosphorylation motifs. (C) A plot showing the relative abundance of amino acids flanking a phosphorylated serine (S) and threonine (T) using the intensity map. Red indicated that this amino acid was significantly enriched near the modification site, and green indicated that this amino acid was significantly reduced near the modification site. Letters represent abbreviations for amino acids.

'DNA conformation change' (Fig. 6A); 'nucleosome' (Fig. 6B), and 'transporter activity' (Fig. 6C), respectively.

To reveal the metabolic pathways involved in the formation of brown films induced by blue light, the DPPs were further analyzed using the KEGG database. For the up-regulated DPPs, two KEGG pathways, 'Ribosome biogenesis in eukaryotes', and 'ABC transporters', were significantly enriched' (Fig. 7A). For the down-regulated DPPs, four enriched KEGG pathways were identified, 'Valine, leucine and isoleucine degradation', 'Phenylalanine metabolism', 'Galactose metabolism', and 'Fructose and mannose metabolism' (Fig. 7B). We also found that the total polysaccharides of blue light treatment was significantly lower than that of red light treatment (Fig. S1). A protein domain enrichment analysis revealed that the up-regulated DPPs were enriched in 19 protein domains, with 'ABC transporter-like', 'P-type ATPase', and 'HAD-like domain' being the most highly enriched(Fig. 7C).

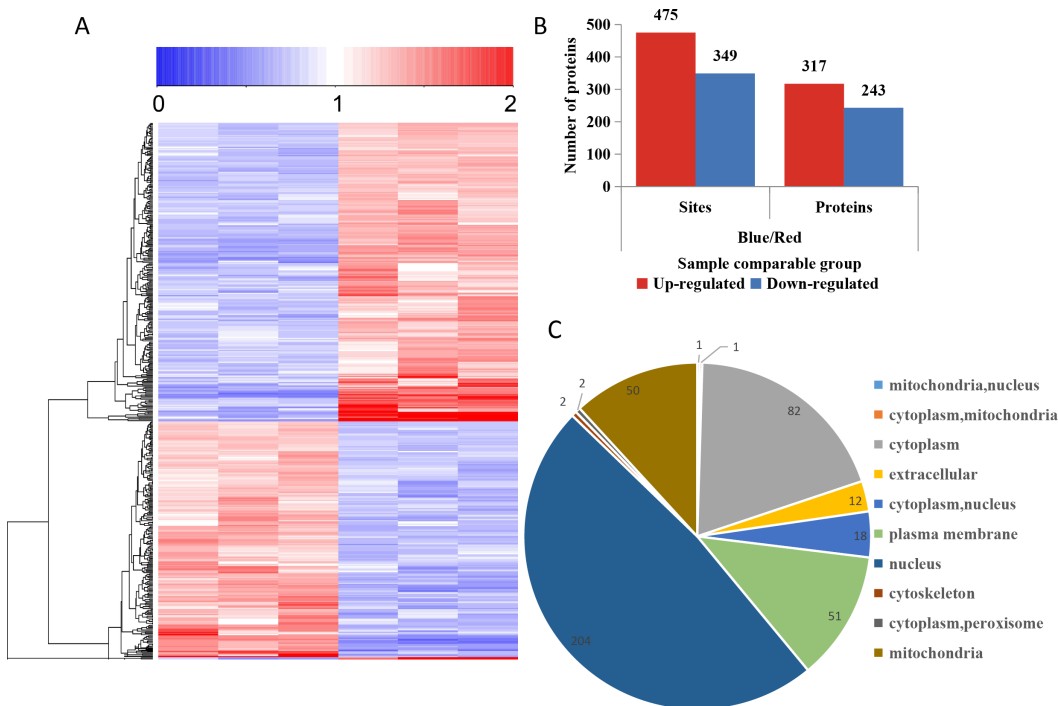

**Figure 5** **Impacts of illumination treatment on phosphorylation proteome levels in fungal mycelia.**
(A) Heat map for the accumulation levels of all the identified phosphorylated proteins. Red indicates up-regulation and green indicates down-regulation. The heatmap scale ranges from 0 to +2. (B) The numbers of up- and down-regulated sites and proteins in red and blue light treatment comparison. (C) Subcellular locations of differentially phosphorylated proteins.

The down-regulated DPPs were most strongly associated with 'Glutathione S-transferase, C-terminal-like', 'YTH domain' 'VPS9 domain', 'Domain of unknown function DUF1708', and 'High mobility group box domain'(Fig. 7D).

## Identification of DPPs related to signal transduction mechanisms and carbohydrate-active enzymes (CAZymes)

To better understand the DPPs related to blue light-induced mycelial brown film formation, a functional classification of DPPs was conducted using euKaryotic Ortholog Groups. A total of 319 DDPs were grouped into 23 subcategories (Fig. S2). For the 'signal transduction mechanisms' subcategory, 50 phosphosites in 29 phosphorylated proteins were identified (Table 1). Among these, 30 phosphosites were up-regulated and 20 were down-regulated.

CAZymes, including auxiliary activity (AA), carbohydrate-binding modules (CBM), carbohydrate esterase (CE), glycoside hydrolase (GH), glycosyl transferase (GT), and polysaccharide lyase (PL), were involved in the hydrolysis of plant cell wall polysaccharides and play an important role in the degradation of substrates(*Davies & Williams, 2016*). In the present study, 13 DPPs were identified as CAZymes, including 11 phosphosites in three CBMs, two phosphosites in two CEs, four phosphosites in three GHs, and six phosphosites in five GTs (Table 2). Interestingly, the GHs were up-regulated, while the CBMs were down-regulated.

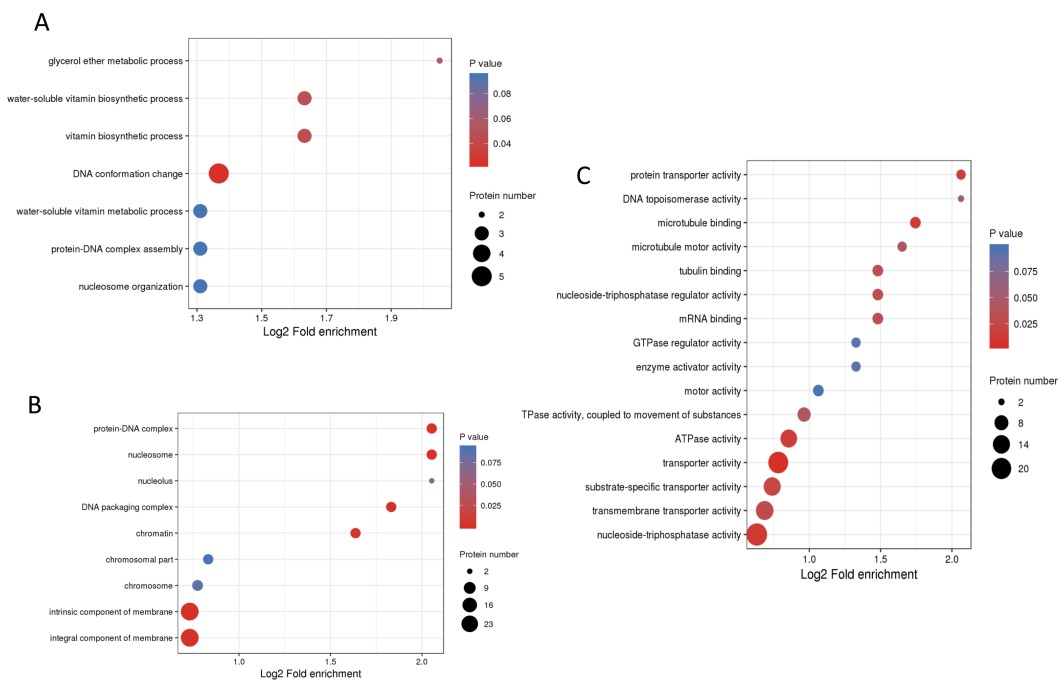

**Figure 6** GO enrichment analysis of DPPs based on biological process (A), cellular component (B) and molecular function (C).

## DISCUSSION

With the completion of various biological genome sequences, proteomics has become an increasingly important analysis of important proteins based on the differential recognition of their expression levels. Protein phosphorylation is an important PTM, which can rapidly control enzyme activity, subcellular localization, and protein stability, and involves the regulation of metabolism, transcription, and translation, as well as protein degradation, homeostasis, cell signaling, and communication (*Lv et al., 2014*; *Thingholm, Jensen & Larsen, 2009*; *Yu et al., 2019*). Recently, large-scale quantitative phosphoproteomics analyses were performed in many plants to elucidate the growth, development, and diverse response mechanisms, but the technology has rarely been applied to *L. edodes* (*Lv et al., 2014*). Here, we report a comprehensive analysis of phosphoproteomic responses to blue light-induced mycelial brown film formation of *L. edodes* through a combination of affinity enrichment and LC-MS/MS.

Protein phosphorylation is a common PTM, but the level of phosphorylation varies with species., The number of phosphorylation sites in each protein is 3.22, which is higher than most published phosphorylation proteomes, indicating that the degree of phosphorylation in the *L. edodes* proteome is very high. The large number of identified phosphoproteins provide an opportunity to comprehensively analyze the mechanism of blue light-induced mycelial brown film formation. The 'sP' motif most frequently occurred in many species, including *L. edodes* (*Van Wijk et al., 2014*; *Wang et al., 2014*; *Zhang et al., 2014*). 'sP' is a target of the following kinases: cyclin-dependent kinase, mitogen-activated protein kinase

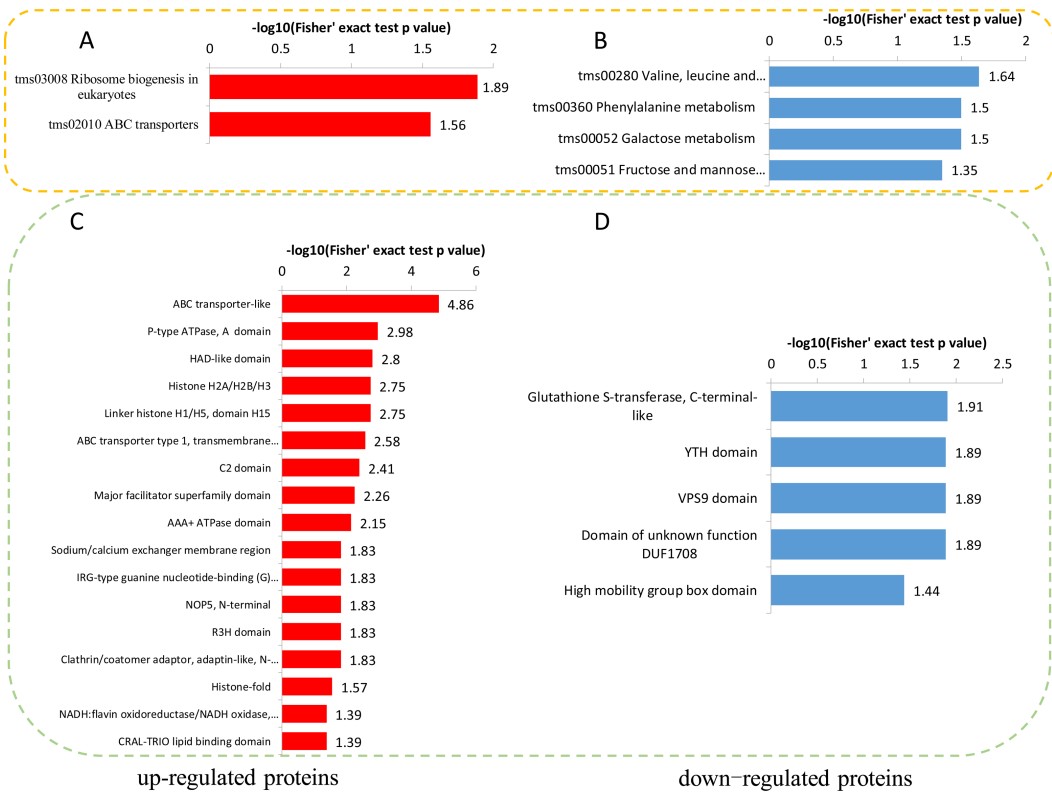

**Figure 7** **KEGG and domain enrichment analysis of the DPPs in fungal mycelium between two different illumination treatments.** KEGG enrichment analysis of up- (A) and down-regulated (B) phosphorylated proteins. Protein domains enrichment analysis of up- (C) and down-regulated (D) phosphorylated proteins.

(MAPK), and sucrose non-fermenting1-related protein kinase 2 (*Van Wijk et al., 2014*; *Zhang et al., 2014*). The 'tP'motif also provides a target for MAPKs (*Wang et al., 2013*).

In Basidiomycetes, light is a crucial environmental factor that affects fruiting body induction and development (*Kues, 2000*; *Kues & Liu, 2000*). In recent years, in fungi, the effects of different light wavelengths on mycelial morphology, metabolites, and enzymatic activities have been studied. In Monascus, red and blue light can affect the formation of mycelia and spores, as well as the production of secondary metabolites (*Miyake et al., 2005*). In this study, we found that blue light can promote the formation of a brown film associated with *L. edodes* mycelia, but no correlation was found with a red-light treatment. The effects of blue light on the expression levels of phosphorylated proteins during brown film formation were studied. Phosphorylation proteomics revealed that 560 phosphorylated proteins were differentially expressed during a blue-light treatment.

Brown film formation at the transcriptional level is correlated with photoreceptor activity, light signaling pathways, and pigment formation (*Tang et al., 2013*). Most fungi perceive blue light through homologues of the white collar complex, which is a complex of photoreceptors and transcription factors that was first found in Neurospora crassa (*Tagua et al., 2015*). The N-terminus of WC-1 is a lov domain, which is a special Per-Arnt-Sim (PAS)

**Table 1  List of differentially expressed signal transduction mechanisms related phosphosites.**

| Protein accession | Position | Ratio | Regulated Type | P value | Amino acid | Protein description |
|---|---|---|---|---|---|---|
| A0A1Q3DXT2 | 147 | 2.041 | Up | 0.000701 | S | Actin cytoskeleton-regulatory complex protein pan1 |
| A0A1Q3DXT2 | 940 | 0.661 | Down | 0.0212 | S | Actin cytoskeleton-regulatory complex protein pan1 |
| A0A1Q3DXT2 | 340 | 1.566 | Up | 0.00104 | S | Actin cytoskeleton-regulatory complex protein pan1 |
| A0A1Q3DXT2 | 538 | 1.503 | Up | 0.000622 | S | Actin cytoskeleton-regulatory complex protein pan1 |
| A0A1Q3DXT2 | 145 | 2.041 | Up | 0.000701 | S | Actin cytoskeleton-regulatory complex protein pan1 |
| A0A1Q3DXT2 | 626 | 1.573 | Up | 0.00602 | S | Actin cytoskeleton-regulatory complex protein pan1 |
| A0A1Q3EQA0 | 342 | 0.666 | Down | 0.0269 | S | Arf gtpase activator |
| A0A1Q3EH65 | 144 | 0.457 | Down | 0.00398 | S | Carbohydrate-binding module family 21 protein |
| A0A1Q3EH65 | 184 | 0.46 | Down | 0.000337 | S | Carbohydrate-binding module family 21 protein |
| A0A1Q3EH65 | 1,114 | 0.634 | Down | 0.00106 | S | Carbohydrate-binding module family 21 protein |
| A0A1Q3EH65 | 391 | 0.512 | Down | 0.0000778 | S | Carbohydrate-binding module family 21 protein |
| A0A1Q3EIZ3 | 287 | 0.542 | Down | 0.0000823 | S | Casein kinase II subunit beta |
| A0A1Q3EIZ3 | 363 | 0.472 | Down | 0.000779 | S | Casein kinase II subunit beta |
| A0A1Q3EHC7 | 350 | 1.787 | Up | 0.0024 | S | Ck1 ck1 ck1-d protein kinase |
| A0A1Q3EML6 | 60 | 0.596 | Down | 0.00757 | S | Gtpase-activating protein gyp7 |
| A0A1Q3DYV9 | 5 | 1.542 | Up | 0.00764 | S | Guanine nucleotide-binding protein |
| A0A1Q3DYV9 | 220 | 1.778 | Up | 0.000163 | S | Guanine nucleotide-binding protein |
| A0A1Q3EBC7 | 120 | 0.663 | Down | 0.00414 | T | HCP-like protein |
| A0A1Q3EQ51 | 108 | 2.293 | Up | 0.0123 | S | Kinase-like protein |
| A0A1Q3EEF5 | 191 | 0.653 | Down | 8.16E−07 | Y | Map kinase |
| A0A1Q3EEF5 | 189 | 0.641 | Down | 1.16E−06 | T | Map kinase |
| A0A1Q3EEF5 | 194 | 0.572 | Down | 0.0000373 | T | Map kinase |
| A0A1Q3E4D7 | 4 | 0.566 | Down | 0.000739 | Y | Mitogen activated protein kinase-like protein |
| A0A1Q3EII7 | 44 | 0.59 | Down | 0.00196 | Y | mRNA stability protein OS=Lentinula edodes |
| A0A1Q3E829 | 28 | 0.651 | Down | 0.000319 | Y | Neutral alkaline nonlysosomal ceramidase |
| A0A1Q3EKW8 | 21 | 1.828 | Up | 0.00182 | S | Non-specific serine/threonine protein kinase |
| A0A1Q3EKW8 | 19 | 1.717 | Up | 0.0000797 | S | Non-specific serine/threonine protein kinase |
| A0A1Q3E982 | 790 | 0.53 | Down | 0.023 | S | Non-specific serine/threonine protein kinase |
| A0A1Q3E1S4 | 101 | 0.642 | Down | 0.00332 | S | Otu-like cysteine |
| A0A1Q3E102 | 696 | 1.667 | Up | 0.0108 | S | Phosphatidylinositol 3-kinase VPS34 |
| A0A1Q3E326 | 400 | 0.496 | Down | 0.000677 | S | Protein phosphatase 2c |
| A0A1Q3E326 | 588 | 0.554 | Down | 0.0000166 | S | Protein phosphatase 2c |
| A0A1Q3E326 | 402 | 0.5 | Down | 0.0000243 | T | Protein phosphatase 2c |
| A0A1Q3E326 | 586 | 0.51 | Down | 0.0000438 | S | Protein phosphatase 2c |
| A0A1Q3E326 | 393 | 0.655 | Down | 0.00102 | S | Protein phosphatase 2c |
| A0A1Q3EFR0 | 271 | 0.543 | Down | 0.000024 | S | Protein serine threonine phosphatase 2C |
| A0A1Q3E8Y4 | 138 | 1.593 | Up | 0.000198 | S | Ras guanyl-nucleotide exchange factor |
| A0A1Q3EIH9 | 765 | 0.636 | Down | 0.000277 | S | Rho gtpase activator |
| A0A1Q3DW25 | 452 | 0.335 | Down | 0.000459 | S | Serine threonine-protein kinase |
| A0A1Q3E8M7 | 137 | 0.648 | Down | 0.00158 | S | SGS-domain-containing protein |

**Table 1** (*continued*)

| Protein accession | Position | Ratio | Regulated Type | *P* value | Amino acid | Protein description |
|---|---|---|---|---|---|---|
| A0A1Q3EKV3 | 265 | 1.703 | Up | 0.039 | S | Signal transducer |
| A0A1Q3EKV3 | 267 | 2.602 | Up | 0.000018 | Y | Signal transducer |
| A0A1Q3EKV3 | 263 | 2.391 | Up | 9.39E−07 | S | Signal transducer OS=Lentinula edodes |
| A0A1Q3E1B1 | 1413 | 1.776 | Up | 0.0000573 | S | Sin component scaffold protein cdc11 |
| A0A1Q3DX25 | 818 | 0.602 | Down | 0.0273 | S | TKL TKL-ccin protein kinase |
| A0A1Q3EHP8 | 698 | 2.269 | Up | 0.000503 | S | Uncharacterized protein |
| A0A1Q3ECY7 | 1,099 | 1.628 | Up | 0.000258 | S | Uncharacterized protein |
| A0A1Q3EHP8 | 695 | 2.269 | Up | 0.000503 | S | Uncharacterized protein |
| A0A1Q3E1M6 | 515 | 0.586 | Down | 0.00232 | S | YTH domain-containing protein 1 |
| A0A1Q3E1M6 | 531 | 0.639 | Down | 0.0227 | S | YTH domain-containing protein 1 |

**Table 2** List of differentially expressed carbohydrateactive enzymes family phosphosites.

| Protein accession | Position | Ratio | Regulated Type | *P* value | Amino acid | Protein description |
|---|---|---|---|---|---|---|
| glycoside hydrolase | | | | | | |
| A0A1Q3DVW4 | 888 | 1.522 | Up | 0.00618 | S | Glycoside hydrolase family 105 protein |
| A0A1Q3DVY0 | 489 | 1.774 | Up | 0.000241 | S | Glycoside hydrolase family 1 protein |
| A0A1Q3DVY0 | 481 | 1.508 | Up | 0.0398 | S | Glycoside hydrolase family 1 protein |
| A0A1Q3EE19 | 429 | 1.931 | Up | 0.0000401 | S | Glycoside hydrolase family 61 protein |
| carbohydrate-binding module | | | | | | |
| A0A1Q3DXJ6 | 394 | 0.555 | Down | 0.0000212 | T | Carbohydrate-binding module family 48 |
| A0A1Q3DXJ6 | 396 | 0.556 | Down | 0.000962 | T | Carbohydrate-binding module family 48 |
| A0A1Q3DXJ6 | 377 | 0.483 | Down | 0.00000158 | S | Carbohydrate-binding module family 48 |
| A0A1Q3DXJ6 | 409 | 0.503 | Down | 0.0000161 | S | Carbohydrate-binding module family 48 |
| A0A1Q3DXJ6 | 380 | 0.508 | Down | 0.0000025 | S | Carbohydrate-binding module family 48 |
| A0A1Q3DXJ6 | 388 | 0.451 | Down | 0.0000568 | S | Carbohydrate-binding module family 48 |
| A0A1Q3E7W8 | 134 | 0.561 | Down | 0.0000969 | S | Carbohydrate-binding module family 12 |
| A0A1Q3EH65 | 1,114 | 0.634 | Down | 0.00106 | S | Carbohydrate-binding module family 21 |
| A0A1Q3EH65 | 391 | 0.512 | Down | 0.0000778 | S | Carbohydrate-binding module family 21 |
| A0A1Q3EH65 | 184 | 0.46 | Down | 0.000337 | S | Carbohydrate-binding module family 21 |
| A0A1Q3EH65 | 144 | 0.457 | Down | 0.00398 | S | Carbohydrate-binding module family 21 |
| carbohydrate esterase | | | | | | |
| A0A1Q3E195 | 171 | 0.312 | Down | 0.000639 | S | Lipase from carbohydrate esterase family ce10 |
| A0A1Q3EGY1 | 39 | 0.605 | Down | 0.000116 | T | Lipase from carbohydrate esterase family ce10 |
| glycosyl transferase | | | | | | |
| A0A1Q3DXW9 | 1,240 | 1.663 | Up | 0.000342 | S | Glycosyltransferase family 20 protein |
| A0A1Q3E591 | 146 | 0.655 | Down | 0.00272 | T | Glycosyltransferase family 4 protein |
| A0A1Q3E591 | 24 | 2.532 | Up | 0.0291 | S | Glycosyltransferase family 4 protein |
| A0A1Q3EH60 | 1,581 | 1.585 | Up | 0.000836 | S | Glycosyltransferase family 2 protein |
| A0A1Q3EI36 | 235 | 0.64 | Down | 0.009 | S | Glycosyltransferase Family 22 protein |
| A0A1Q3ERC2 | 75 | 0.622 | Down | 0.00812 | T | Glycosyltransferase family 2 protein |

domain that can bind to flavin adenine dinucleotide (*Crosson, Rajagopal & Moffat, 2003*). Light sensing via photoreceptors such as FMN- and FAD-bindings and signal transduction by kinases and G protein-coupled receptors were identified as differential expression genes specific to the light-induced brown film phenotypes (*Yoo et al., 2019*). In the present study, three flavin adenine dinucleotide-binding domains and an FMN-binding domain differentially accumulated, indicating that the *L. edodes* mycelia could have perceived blue light when the brown film was formed. The MAPK cascade is an important signal transduction pathway connecting light responses and the biological clock (*De Paula et al., 2008*). MAPK also regulates various secondary metabolic activities in *Aspergillus nidulans* and *Colletotrihum lagenarium*, and it controls light-influenced melanin biosynthesis in *B. cinerea* (*Atoui et al., 2008*; *Bayram & Braus, 2012*; *Liu et al., 2011*; *Takano et al., 2000*). The MAPK signal transduction pathways may be directly involved in brown film formation (*Tang et al., 2013*). Several MAPK signal transduction pathways related to DPPs were identified in this study, suggesting that these signal pathways are involved in the formation of brown films.

The differential expression of CAZymes were observed in *L. edodes* mycelia under two light conditions. GHs mainly hydrolyze glycosidic bonds between carbohydrates or between carbohydrates and non-carbohydrates (*Sathya & Khan, 2014*). The GH61 family contains copper-dependent lytic polysaccharide monooxygenase (*Langston et al., 2011*). CEs catalyze the deacylation of esters or amides, in which sugar plays the role of alcohol and amine (*Biely, 2012*; *Vidal-Melgosa et al., 2015*). They are currently divided into 16 different families, which have a great diversity in substrate specificity and structure (*Vidal-Melgosa et al., 2015*). CE10 (two DPPs) were down-regulated by blue light. CBMs are noncatalytic, individually folded domains that are attached to the catalytic enzyme modules by linkers (*Varnai et al., 2014*). Some CE1 enzymes may contain a CBM48 family protein, which is associated with starch binding (*Wilkens et al., 2017*; *Wong et al., 2017*). Our research showed that these CAZymes play important roles in the degradation of lignocellulose and provide sufficient nutrition for the formation of the brown film of mushroom mycelia.

To survive, fungi have evolved the ability to adapt to different environmental conditions, and various metabolic pathways secrete different metabolites (*Yu & Keller, 2005*). The regulation of these metabolites is not only related to fungal growth and development, but also to light stimulation and responses. The shorter the light wavelength, the more polysaccharides accumulated in the cells of *Pleurotus eryngii* (*Jang et al., 2011*). Blue-light treatments significantly improved the synthesis of ergosterol and polyphenols in the fruiting body of *Pleurotus eryngii*, and the scavenging ability of the free radicals was the greatest compared with other light treatments (*Jang et al., 2011*). In our study, the KEGG-enrichment analysis showed that four DPPs belonged to 'Galactose metabolism' and 'Fructose and mannose metabolism', suggesting that the blue light affected the sugar metabolism of *L. edodes*. Phenolic compounds were correlated with pigment formation (*Weijn et al., 2013*). Phenylalanine ammonia-lyase and tyrosinase-encoding genes were significantly up-regulated in *P. eryngii* under blue-light conditions (*Du et al., 2019*). Two 'Phenylalanine metabolism' pathway phosphoproteins, amidase (A0A1Q3E9W2) and

aspartate aminotransferase (A0A1Q3EG41), were down-regulated in mycelia under blue-light conditions. These results suggested that blue light may promote the formation of melanin and inhibit the formation of other phenolic compounds. Polyketide synthase (PKS) is an essential enzyme in the biosynthesis of fungal secondary metabolites (*Austin & Noel, 2003*; *Linnemannstons et al., 2002*). PKSs modify the polyketide backbone with other enzymes, such as Cytochrome P450 monooxygenases, oxidoreductase, and omethyltransferase (*Austin & Noel, 2003*). P450-linked monooxygenases mediate oxidation–reduction steps in aflatoxin biosynthesis, and omethyltransferase was involved in yellow pigment biosynthesis through an aflatoxigenic Aspergillus strain (*Bhatnagar, Ehrlich & Cleveland, 2003*). In our study, the phosphorylation levels of PKS, O-methyltransferase, P450 monooxygenase, and oxidoreductase changed in brown film formation, indicating that they may play roles in pigment production.

The ABC transport family is widely distributed in all living species, including several subfamilies, which are responsible for different types of material transport (*Higgins, 2001*; *Holland & Blight, 1999*). ATPase is the largest ATP dependent ion transporter in organisms, transporting many different ions, metals and other substrates (*Palmgren MG. Nissen, 2011*). Two VPS9 domain containing proteins, Rab5 GDP/GTP exchange factor, were down-regulated under blue light treatment. Studies have shown that the transport of endocytic vesicles is partially regulated by Rab protein (*Zhu et al., 2018*). Rab protein needs to be activated by guanine nucleotide exchange factor, which transforms Rab from a GDP binding state to a GTP binding state (*Zerial & McBride, 2001*). The changed in these proteins suggest that blue light altered the transport of certain substances. In mushrooms, blue light can promote growth, which is considered to be an important environmental factor affecting the growth of fruiting bodies (*Yoo et al., 2019*). In this study, ribosome biogenesis related proteins were observed to be up-regulated under blue light treatment.

## CONCLUSIONS

Using a high-resolution LC-MS/MS integrated with a highly sensitive immune-affinity antibody method, phosphoproteomes of *L. edodes* mycelia under red- and blue-light conditions were analyzed. In this study, 11,224 phosphorylation sites were identified on 2,786 proteins, of which 9,243 sites on 2,579 proteins contained quantitative information. In total, 475 sites were up-regulated and 349 sites were down-regulated in the blue vs red group. Then, we carried out a systematic bioinformatics analyses of proteins containing quantitative information sites, including protein annotations, functional classifications, and functional enrichments. Our study provides new insights into the molecular mechanisms of the blue light-induced brown film formation at the PTM level.

## ACKNOWLEDGEMENTS

We are grateful to the PTM Biolabs company for technical support. We thank International Science Editing for editing this manuscript.

### Funding

This work was supported by the Zhejiang Science and Technology Major program on Agriculture New Variety Breeding (Grant No.2016C02057) and National Science Foundation of Zhejiang Province of China (Grant No.LQ16C150004). The funders had no role in study design, data collection and analysis, decision to publish, or preparation of the manuscript.

### Grant Disclosures

The following grant information was disclosed by the authors:
Zhejiang Science and Technology Major program on Agriculture New Variety Breeding: 2016C02057.
National Science Foundation of Zhejiang Province of China: LQ16C150004.

### Competing Interests

The authors declare there are no competing interests.

### Author Contributions

- Tingting Song and Weiming Cai conceived and designed the experiments, performed the experiments, analyzed the data, prepared figures and/or tables, authored or reviewed drafts of the paper, and approved the final draft.
- Yingyue Shen and Qunli Jin performed the experiments, analyzed the data, authored or reviewed drafts of the paper, and approved the final draft.
- Weilin Feng and Lijun Fan performed the experiments, analyzed the data, prepared figures and/or tables, and approved the final draft.

### Data Availability

The mass spectrometry proteomics data are available at the ProteomeXchange Consortium: PXD016536.

### Supplemental Information

Supplemental information for this article can be found online at http://dx.doi.org/10.7717/peerj.9859#supplemental-information.

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
