# Peer review of "Comparative phosphoproteome analysis to identify candidate phosphoproteins involved in blue light-induced brown film formation in Lentinula edodes"

_PeerJ, doi:10.7717/peerj.9859_

## Round 0.1 · original submission · Major Revisions

Dear Dr. Song and colleagues:

Thanks for submitting your manuscript to PeerJ. I have now received three independent reviews of your work, and as you will see, one reviewer recommended rejection, while another suggested a major revision. I am affording you the option of revising your manuscript according to all three reviews but understand that your resubmission may be sent to at least one new reviewer for a fresh assessment (unless the reviewer recommending rejection is willing to re-review).

The methods should be clear, concise and repeatable. Please ensure this, and make sure all relevant information and references are provided. Also, elaborate on the discussion of your findings, placing them within a broad and inclusive body of work by the field (relevance to fungal biology).

Please fix all of the identified issues with figures and tables.

Therefore, I am recommending that you revise your manuscript, accordingly, taking into account all of the issues raised by the reviewers.

I look forward to seeing your revision, and thanks again for submitting your work to PeerJ.

Good luck with your revision,

-joe

·

Basic reporting

Song et al. performed phosphoproteome analysis in L. edodes. This manuscript lacks many important descriptions, which may cause misunderstanding. Also, I could not find any importance in fungal biology. So, I do not recommend to publish this manuscript.

Specific comments
1. No importance in phosphoproteome is stated in the current manuscript, although I also suppose it may be important. Why was comparative/differential phosphoproteome analysis conducted in the two samples? Especially, relationship between blue light and protein phosphorylation was not described. Just performing experiment and analysis using new samples?

2. The authors state that many DPPs were identified. But, I suppose most of them may be proteins differentially expressed between the two samples, but not those of which phosphorylation levels differ. Did the author analyze simple proteome analysis using total proteins from the two samples to normalize? I could not find any descriptions about this in the current manuscript. If not, the manuscript should be rewritten to avoid misunderstanding.

3. The culture condition is not well described. What medium was used in this study? What is the sources of blue and red light? Was the incubator illuminated all day or under cycles such as 12 h ON/ 12 h OFF?

4. Discussion must be rewritten because it does not contain “discussion” based on the results, but many descriptions that could be used for Introduction of the other manuscripts. Almost of the descriptions are not relate to this study and the results. The most serious description is lines 351-354, which is not new finding in this manuscript.


5. I suppose many proteins that exhibit affinity to metal ion are listed, suggesting enrichment of phosphorylated proteins by IMAC may not be enough or background is high.

6. Lines 232-243: Is this description based on the MS analysis or genome database?

Experimental design

The authors state that many DPPs were identified. But, I suppose most of them may be proteins differentially expressed between the two samples, but not those of which phosphorylation levels differ. Did the author analyze simple proteome analysis using total proteins from the two samples to normalize? I could not find any descriptions about this in the current manuscript. If not, the manuscript should be rewritten to avoid misunderstanding.
I suppose many proteins that exhibit affinity to metal ion are listed, suggesting enrichment of phosphorylated proteins by IMAC may not be enough or background is high.

Validity of the findings

I suppose phosphoproteome itself is important.

·

Basic reporting

Song and coworkers have applied a phosphoproteomic approach to identify phosphorylated proteins whose formation is influenced by blue light. The experimental methods used for this purpose seem to have been applied with experience and care. I have, however, several concerns about the bioinformatic approach used to identify the respective phosphoproteins:

Experimental design

As for the "biological" experiments, I have no concerns. They are brief and simple, and the techniques described clearly.

Validity of the findings

As noted above, there are several problems:

1. The resulting MS/MS data were processed using the Maxquant search engine (v.1.5.2.8), and queried against a human Uniprot database concatenated with a reverse decoy database. This is a standard procedure in most studies. However, protein sequences of Human orthologs that can be found in Lentinus edodes are usually poorly conserved, and one would have to be very lucky to find peptides that bear the similarity that the authors claim. The genome of Lentinus edodes has been sequenced (Chen et al. PLoS One. 2016;11(8):e0160336) and it would therefore be possible to retrieve the sequences for the whole putative proteins of all the identified peptides. This would put the putative identification of the phosphopeptides on a much more reliable level. A (supplementary) table that lists these peptides, the respective proteins deduced from the genome sequence, and their accession numbers needs to be presented.

2. I have a similar problem with GO and KEGG analysis: GO is very popular because it enables an enrichment analysis of proteins belonging to the same category. However, GO has not been optimized for multicellular fungi, but is strongly based on yeast, plant and bacterial databases. Consequently, GO analysis always introduces an identification error of 5-10 % with fungal proteins, and I am afraid that this would be particularly the case if only peptides are used. This problem could be solved in a similar way as above: don’t use the peptides but the full length proteins (extracted from the genome sequence), and compare the resulting identity with the results of a blast search and conserved domain analysis. Such a (supplementary) table, including the best hit and its statistics, needs to be presented.

3. As for the KEGG analysis, I request that the authors name the proteins that they claim to be enriched rather than only giving the KEGG group, because some of the groups provide no information. Examples: “fructose metabolism”. Fructose can be metabolized by the glycolytic pathway without any additional enzyme, and I therefore wonder which protein gave rise to this claim of enrichment. Or “galactose metabolism”: I looked up the respective KEGG pathway sheet and found that it lacks the enzymes of the alternative, reducing galactose catabolic pathway that has been identified in fungi. So do the authors mean the Leloir enzymes? I believe this attribution of phosphoproteins to KEGG groups needs to be included in the Supplementary Table I requested above.
I believe these the above three points are essential to present the results of this study in a way that it provides a useful and reliable information to the readers. Yet there are additional points that should be considered too:

Additional comments

The authors detect phosphorylation in glycoside hydrolases and their polysaccharide binding domains. This is new. Phoshorylation has been detected in the carbohydrates of the attached glycan chains of these glycoproteins but never on the proteins themselves. The authors should discuss the potential significance of this finding. Phosphorylation of a protein alters its isoelectric point and consequently the solubility. Would the phosphorylation improve solubility or binding to the substrate? This is a point where a rather simple experiment could have contributed significantly to the paper.

One limitation of the paper is its descriptive nature, and the use of broad categorizations in the present manuscript does not allow any clear insights. It would significantly improve the manuscript if the authors would be able to provide some additional experiments that prove the importance of some of the identified peptides in the response to blue light.

In Table1 (List of differentially expressed signalling proteins) the authors use a cut-off of 1.5-fold. This appears very low to me. In transcriptome analyses, a log2 >2 is the current

Reviewer 3 ·

Basic reporting

Overall there is comprehensive information provided, but still clarity is lacking in few aspects.
Some figures are mislabeled, and also resolution for figures is poor. A couple of them are hard to see/read (details below). In the introduction, authors very briefly touch through the blue light photoreceptors. However, in the context of this manuscript it will be nice to have some more background on their biological roles specifically in context of Lentinula edodes. Also, authors are missing some very recent important references such as Yoo Seung-il et al, BMC Genomics 20, 121 (2019) though on transcriptional study but is highly in context of this particular study. One another reference is Kim J Y et al, PLoS ONE 15(3): e0230680 (2020). These should be briefly discussed. Raw data is provided in the supplementary tables. Language is mostly clear.

Experimental design

Experimental:
Information on experimental replicates is not clear. It seems three biological replicates are taken. Are they take in parallel, or on three different times? Authors should clarify this in text. Also, methodology section needs more clarification. Points mentioned under general comments.

Validity of the findings

No Comment

Additional comments

This is a novel study elucidating differential phosphoproteome involved in mycelial browning of basidiomycete fungi Lentinula edodes. Overall there is comprehensive information provided, but still clarity is lacking in few aspects. More expansion in introduction in needed to strengthen the foundation of study, and in discussion section to justify the key experimental inferences.

1. A better picture with higher resolution will be good for the red vs blue light treatment of the L. edodes. (Fig 1A)
2. Instead of just stating blue or red light, authors should also provide lumen intensity of the lights used in this study.
3. In the material and method section, trypsinization strategy is confusing. Were there two rounds of trypsinization one after another (Lines 147-149).
4. Which TMT kit?, What IMAC spheres were used Fe, Ti,..?? Provide more detailed and accurate description.
5. Include more description on how the phosphosites were identified, and how is the quantification done.
6. Figure 1C, hard to read the axis of the graph
7. Figure 2B, Label axis
8. Figure 3B is mislabeled in the legend, also not clear how is B different from A
9. Figure 4C, provide more description on the intensity map (to clarify the understanding, distance and position are cleat, what does red and green signify in fig 4C, numbers or something else)
10. Legend not accurate. 5C? Also, resolution of figures very poor, hence sometimes difficult to read no's
11. Line 297, correct Fig7B to 7A
12. Did authors see/analyze any change in localization of proteins in red vs blue light treatments? That will be an interesting information which can be easily fished from the given data.
13. Authors, should expand the discussion section adding some key points. A. For DPP, KEGG enrichment analysis (Fig7) up-regulated phosphorylated proteins show ABC transporters, and ribosome biogenesis phosphorylated proteins. authors should put this in discussion, as why they think these are upregulated and possible relevance to brown film formation. B. Expanding on the similar lines, for top upregulated domains, ABC transporter like, P-type ATPase, etc.
Similarly, discussion should be expanded more for downregulated phosphorylated proteins and domains.

---

## Round 0.2 · accepted · Accept

Dear Dr. Song and colleagues:

Thanks for revising your manuscript based on the concerns raised by the reviewers. I now believe that your manuscript is suitable for publication. Congratulations! I look forward to seeing this work in print, and I anticipate it being an important resource for groups studying phosphoproteins involved in blue light-induced brown film formation in Lentinula edodes. Thanks again for choosing PeerJ to publish such important work.

Best,

-joe

·

Basic reporting

The MS has been modified as I had required.

Experimental design

Non

Validity of the findings

Non

Additional comments

Non

Reviewer 3 ·

Basic reporting

Manuscript is improved, and basic reporting is fine.

Experimental design

Explained better

Validity of the findings

Fine

Additional comments

Revised manuscript has addressed most of the concerns. Figures, seemed to be improved. I recommend accepting the article. Though text reading for minor text mistakes is needed.